# Critical Aspects Concerning the Development of a Pooling Approach for SARS-CoV-2 Diagnosis Using Large-Scale PCR Testing

**DOI:** 10.3390/v13050902

**Published:** 2021-05-13

**Authors:** Daniel Cruceriu, Oana Baldasici, Loredana Balacescu, Stefana Gligor-Popa, Mirela Flonta, Milena A. Man, Simona Visan, Catalin Vlad, Adrian P. Trifa, Ovidiu Balacescu, Patriciu Achimas-Cadariu

**Affiliations:** 1Department of Genetics, Genomics and Experimental Pathology, The Oncology Institute “Prof. Dr. Ion Chiricuta”, 34-36, Republicii Street, 400015 Cluj-Napoca, Romania; dani_cruceriu@yahoo.com (D.C.); oana_baldasici@yahoo.ro (O.B.); loredana_balacescu@yahoo.com (L.B.); simona.visan19@gmail.com (S.V.); 2Department of Molecular Biology and Biotechnology, “Babes-Bolyai” University, 1, M. Kogălniceanu Street, 400084 Cluj-Napoca, Romania; 3Department of Oncogenetics, The Oncology Institute “Prof. Dr. Ion Chiricuta”, 34-36, Republicii Street, 400015 Cluj-Napoca, Romania; stefiipopa@yahoo.com; 4Laboratory of Microbiology, Clinical Hospital of Infectious Diseases, 23, Iuliu Moldovan Street, 400348 Cluj-Napoca, Romania; mflonta4@gmail.com; 5Department of Medical Sciences—Pulmonology, “Iuliu Hatieganu” University of Medicine and Pharmacy, 8, Victor Babes Street, 400012 Cluj-Napoca, Romania; manmilena50@yahoo.com; 6Covid Department, “Leon Daniello” Clinical Hospital of Pulmonology, 6, Hasdeu Street, 400371 Cluj-Napoca, Romania; 7Department of Surgery, The Oncology Institute “Prof. Dr. Ion Chiricuta”, 34-36, Republicii Street, 400015 Cluj-Napoca, Romania; catalinvlad@yahoo.it (C.V.); patrick.achimas@hotmail.com (P.A.-C.); 8Department of Oncology, “Iuliu Hatieganu” University of Medicine and Pharmacy, 8, Victor Babes Street, 400012 Cluj-Napoca, Romania; 9Department of Genetics, “Iuliu Hatieganu” University of Medicine and Pharmacy, 8, Victor Babes Street, 400012 Cluj-Napoca, Romania; 10Department of Medical Oncology, “Iuliu Hatieganu” University of Medicine and Pharmacy, 8, Victor Babes Street, 400012 Cluj-Napoca, Romania

**Keywords:** COVID-19, SARS-CoV-2, sample pooling, molecular diagnostic, nasopharyngeal swabs, RNA extraction, RT-qPCR

## Abstract

The primary approach to controlling the spread of the pandemic SARS-CoV-2 is to diagnose and isolate the infected people quickly. Our paper aimed to investigate the efficiency and the reliability of a hierarchical pooling approach for large-scale PCR testing for SARS-CoV-2 diagnosis. To identify the best conditions for the pooling approach for SARS-CoV-2 diagnosis by RT-qPCR, we investigated four manual methods for both RNA extraction and PCR assessment targeting one or more of the RdRp, N, S, and ORF1a genes, by using two PCR devices and an automated flux for SARS-CoV-2 detection. We determined the most efficient and accurate diagnostic assay, taking into account multiple parameters. The optimal pool size calculation included the prevalence of SARS-CoV-2, the assay sensitivity of 95%, an assay specificity of 100%, and a range of pool sizes of 5 to 15 samples. Our investigation revealed that the most efficient and accurate procedure for detecting the SARS-CoV-2 has a detection limit of 2.5 copies/PCR reaction. This pooling approach proved to be efficient and accurate in detecting SARS-CoV-2 for all samples with individual quantification cycle (Cq) values lower than 35, accounting for more than 94% of all positive specimens. Our data could serve as a comprehensive practical guide for SARS-CoV-2 diagnostic centers planning to address such a pooling strategy.

## 1. Introduction

Despite significant advances in the development and use of COVID-19 vaccines, the pandemic caused by the SARS-CoV-2 coronavirus continues to put pressure on healthcare systems worldwide, given the growing number of new infections and severe cases requiring hospitalization. The primary approach to controlling the spread of COVID-19 disease is to quickly diagnose and isolate the infected people, simultaneously applying protection measures and physical distancing [1]. At this moment, the diagnostic capacity of the national SARS-CoV-2 laboratory in Romania is around 50,000 RT-qPCR tests/day. All diagnostic tests are performed in specialized laboratories in national, regional, and local hospitals or medical centers, in both the public and private sectors of the Romanian healthcare system. However, the daily number of SARS-CoV-2 tests performed in Romania is less than the maximum diagnostic capacity, being around 38,000 tests/day, due to the current COVID-19 surveillance case definition and testing recommendations.

Therefore, a key factor for implementing such a strategy is an efficient, large-scale diagnosis of SARS-CoV-2 infection, especially if one takes into account that a significant proportion (40–78%) of the infected population is asymptomatic [2,3].

The golden standard for COVID-19 diagnosis recommended by the World Health Organization consists in detecting viral SARS-CoV-2 RNA from nasopharyngeal swab specimens by real-time qPCR (RT-qPCR) assessment [4]. However, the diagnostic capacity for COVID-19 disease is limited in many countries by the finite resources available. Thus, too few people are getting tested compared with the real needs [5]. In an attempt to increase the number of tests and cover the need for rapid results, rapid antigen detection kits have been recently developed [6]. However, rapid testing should be performed only by specialized personnel. Otherwise, inadequate collection of the biological sample (nasopharyngeal exudate) may result in false-negative results.

In this context, several recent publications suggest that large-scale group testing, including sample pooling strategies, might improve the efficiency of COVID-19 diagnosis [7,8]. Sample pooling implies mixing several samples and testing them as a single pool, subsequent individual tests being performed only if the pool tests positive. This testing strategy was previously used for other viral diseases, such as AIDS and hepatitis B and C [9].

If proved efficient, such a pooling approach could be easily implemented as a reliable method to detect SARS-CoV-2 for large population groups, extending the diagnostic capacity of many available laboratories. Furthermore, this pooling strategy could be easily applied to large target groups sequentially at certain time intervals (e.g., 7–12 days), identifying the positive people even from the asymptomatic phase. Schools, universities, factories, and companies, all representing large communities, would highly benefit from such a large-scale repetitive testing algorithm, securing their proper functioning. However, the greatest challenge in achieving this goal is maintaining the performance of the diagnostic assay, even after sample pooling, and thus, minimizing the possibility of occurrence of false-negative results.

Recent studies demonstrate that SARS-CoV-2 detection in pooled samples, mixed either before the extraction step [10,11] or before the RT-qPCR reaction [12,13], might be reliable. However, the accuracy of such a pooling testing approach is highly dependent on both viral RNA extraction and its PCR detection procedures. In this context, considering our experience in diagnosing over 40,000 patients for SARS-CoV-2 infection as part of the Romanian laboratory network involved in the national screening program, our paper aims to present critical data concerning pooled samples testing to analyze SARS-CoV-2. Moreover, our data could serve as a comprehensive practical guide for SARS-CoV-2 diagnostic centers planning to address such a pooling strategy.

## 2. Materials and Methods

Our study’s design to establish a reliable method for large-scale testing of SARS-CoV-2 coronavirus by RT-qPCR included four manual methods for RNA extraction and PCR assessment, two PCR devices, and an automated flux for SARS-CoV-2 detection (Figure 1).

### 2.1. Patients and Sample Collection

The nasopharyngeal swab specimens from 35 SARS-CoV-2-positive and 102 SARS-CoV-2-negative patients used in this study were collected between 15 April and 15 May 2020. Nasopharyngeal samples were collected with cotton swabs in a 3 mL viral transport medium (ViroSan Transport Medium, SaniMed, Romania) and stored at 4 °C for no longer than 48 h before RNA extraction. This study was an observational one and not an interventional one; the nasopharyngeal samples harvested from the patients were anonymized and used as usual, without including any personal data of the tested patients, our study being in accordance with the Declaration of Helsinki.

### 2.2. SARS-CoV-2 RNA Extraction

The RNA extraction procedure was manually performed with four different commercially available kits: PureLink Viral RNA/DNA Mini Kit (#12280050, Thermo Fisher Scientific, Waltham, MA, USA), termed THERMO in this article; EliGene Viral DNA/RNA Isolation Kit (#409100, Elisabeth Pharmacon, Brno, Czech Republic), termed ELI in this article; NucleoSpin Dx Virus (#740895.50, Macherey-Nagel, Düren, Germany), termed MN in this article; and Quick-RNA Viral Kit (#R1035, Zymo Research, Irvine, CA, USA), termed ZYMO in this article. Additionally, SARS-CoV-2 RNA was also extracted and detected by an automated procedure using the NeuMoDx SARS-CoV-2 Assay (#300800, NeuMoDx, Ann Arbor, MI, USA), termed NeuMoDx in this article.

The viral RNA was extracted starting from 200 µL of the sample (≥500 µL for the NeuMoDx procedure). All the manual RNA extraction kits are filter spin column based, consisting in four successive steps: sample lysis, RNA binding to the filter membrane, RNA washing, and RNA elution. The final elution volume used was 50 µL. However, because both THERMO and ZYMO extraction kits recommend a final elution volume between 20 and 50 µL, we used both 20 µL and 50 µL elution volumes for these kits. RNA extraction was employed both on nasopharyngeal swab specimens from patients and on a SARS-CoV-2 Reference Material (AccuPlex SARS-CoV-2 Reference Material Kit, #0505-0126, Sera-Care, Milford, MA, USA). At each stage of the experiments, all samples were extracted at the same time.

### 2.3. SARS-CoV-2 Detection by RT-qPCR Amplification

SARS-CoV-2 detection by RT-qPCR was employed with four different commercially available kits: TaqMan SARS-CoV-2 Assay Kit (#CCU002NR) combined with TaqPath 1-Step Multiplex Master Mix (#A28523, Applied BioSystems by Thermo Fisher Scientific, USA), termed THERMO in this article; EliGene COVID19 BASIC A RT Kit (#90077-RT-A, Elisabeth Pharmacon, Czech Republic), termed ELI in this article; Coronavirus (COVID-19) Genesig Real-Time PCR assay (#Z-Path-COVID-19-CE, Primer Design, UK), termed PD in this article; and LightMix Modular SARS-CoV-2 (COVID19) RdRP (#53-0777-96, TIB MOLBIOL, Berlin, Germany), termed TIB in this article.

The THERMO amplification kit separately detects regions from 3 SARS-CoV-2 genes (N protein, ORF1a, and S protein). The Cq used in this study for the THERMO amplification is the average Cq for all three sequences. All the other amplification kits return only one Cq value/sample, corresponding to the amplification of the RdRp gene for the PD and TIB kits and to the N gene (3 sequences) for the ELI amplification kit. Therefore, this unique Cq value was taken into account for these three amplification kits. NeuMoDx amplifies separately two specific targets, N and Nsp2; thus, the Cq value used in this study for the NeuMoDx amplification is the average Cq for both sequences.

PCR amplifications were conducted according to each manufacturer’s protocol. Each PCR run included both an amplification positive control and an RNA extraction control, provided by each manufacturer. A sample was considered positive, according to each manufacturer’s guidelines, as follows: a Cq value lower than 39 for the TIB kit, a Cq value lower than 40 for the ELI kit, and a Cq value lower than 40 for any two out of the three genes tested for the THERMO kit and all instances of test sample amplification for the PD kit. Two RT-qPCR instruments were used in this study: LightCycler480 (Roche, Basel, Switzerland) and QuantStudio5 (Applied Biosystems by Thermo Fisher Scientific, Waltham, MA, USA). At each stage of the experiments, all samples were amplified in the same RT-qPCR run.

### 2.4. Assessment of the Optimal Pool Size

The optimal pool size was assessed based on the prevalence of COVID-19 disease registered in April 2020 at the Oncology Institute “Prof. Dr. Ion Chiricuta,” Cluj-Napoca, Romania. At that time, a screening for SARS-CoV-2 infection among 471 asymptomatic oncological patients and 1117 asymptomatic medical staff was implemented in the Department of Genetics, Genomics and Experimental Pathology of our Institute. This screening was performed by individual testing.

Based on the observed infection prevalence for each cohort, the optimal pool size was calculated for three different pooling strategies: Dorfman hierarchical testing (2 stages) [14], Sterett hierarchical testing (3 stages) [15] and an array testing approach. All calculations were performed in “The Shiny” application for pooled testing, available at https://www.chrisbilder.com/shiny (accessed on 20 May 2020). The parameters used in this calculation included, besides the experimental prevalence rate, an assay sensitivity of 95%, an assay specificity of 100%, and a range of pool sizes of 3 to 40 samples [10].

### 2.5. Pooling of Nasopharyngeal Swab Specimens

A total of 15 SARS-CoV-2-positive samples were used for pooling. Each positive nasopharyngeal swab specimen (200 µL) was mixed with 4, 9, and 14 negative nasopharyngeal swab samples (200 µL each), obtaining pools of 1 mL (1:5), 2 mL (1:10), and 3 mL (1:15), respectively. Subsequently, 200 µL of each pool was used for extraction and further amplification, as previously described.

## 3. Results

### 3.1. The Relative Efficiency of Different Extraction-Amplification Procedures for SARS-CoV-2 Detection by RT-qPCR

In order to identify the most efficient extraction–-amplification procedure for SARS-CoV-2 detection, congruent with the resources and infrastructure available in our laboratory, multiple components and parameters were taken into account: the extraction kit, the RNA elution volume, the amplification kit, the RT-qPCR instrument, and the overall procedure in terms of manual vs. automated extraction and detection. With this purpose in mind, we selected seven COVID-19-positive patients based on their SARS-CoV-2 $\bar{\mathrm{Cq}}$ values, which varied between 19.3 and 36.8 ($\bar{\mathrm{Cq}}$—an average of all individual Cq obtained for one sample by different extraction–amplification combinations). Thus, patients with both very high and very low initial viral loads were included in this study.

Firstly, all possible combinations (4 × 4) of extraction and amplification kits were evaluated in terms of their relative amplification efficiency, starting from 200 µL of sample and using a final elution volume of 50 µL, for all seven patients. For all samples, the THERMO–THERMO combination of extraction–amplification kits had the highest relative amplification efficiency, whereas the amplification with TIB coupled with the extraction with either ZYMO or MN proved to be the least efficient in detecting SARS-CoV-2 (Figure 2A). Furthermore, for samples with very high $\bar{\mathrm{Cq}}$ values, such as those corresponding to patient 6 (Figure 2A) or 7, several combinations of extraction–amplification kits did not detect SARS-CoV-2 sequences at all (PA7 − $\bar{\mathrm{Cq}}$ = 36.8, detected only by the combinations THERMO-THERMO/ELI/PD and ELI-THERMO (extraction–amplification).

In order to establish if the differences observed between the four extraction kits are significant, a *t*-test was employed for each biological sample by comparing the four Cq values obtained by using one extraction kit vs. the four Cq values obtained by using another extraction kit. The four Cq values corresponding to each extraction kit were obtained with each of the four amplification kits used; thus a paired *t*-test was used. In this analysis, the THERMO extraction kit proved to be statistically significantly superior to all the other extraction kits, with fold changes varying between 1.79 (PA3) and 8.35 (PA6) (Figure 2B). We implemented the same type of data analysis to compare the amplification kits. In this case, the PCR THERMO kit was more efficient in detecting SARS-CoV-2 than all the other amplification kits, with fold changes as high as 9.82 (PA1) (Figure 2C).

Because both THERMO and ZYMO extraction kits recommend an elution volume between 20 and 50 µL, the next step was to evaluate the final elution volume’s impact on the extraction efficiency. Our data indicate that a final elution in 20 µL instead of 50 µL significantly decreased the Cq values and improved the detection efficiency of SARS-CoV-2, independent of the amplification kit used (Figure 2D–F). The average difference in the Cq values between the elution volume of 20 µL and of 50 µL for all samples was 2.4 for samples extracted with ZYMO and amplified with ELI (*p* ** = 0.004), 2.1 for samples extracted with ZYMO and amplified with THERMO (*p* * = 0.018), and 1.0 for samples extracted with THERMO and amplified with THERMO (*p* = 0.11).

By integrating these data back to analyze the efficacy of the extraction kits, we conclude that the RNA extraction with THERMO in a final volume of 20 µL becomes the most efficient option. In comparison, the extraction with ZYMO in 20 µL is as efficient as the one with the ELI extraction kit (Figure 3A).

The last component of the manual extraction–amplification procedure evaluated in this study was the amplification efficiency of two RT-qPCR instruments: LightCycler480 (Roche, Basel, Switzerland) and QuantStudio5 (Thermo Fisher Scientific, Waltham, MA, USA). No significant differences were observed in SARS-CoV-2 detection between the two devices, independent of the extraction or amplification kits used (Figure 2G).

Based on these data, the best yielding SARS-CoV-2 RNA extraction kits appear to be THEMO and ZYMO in a final elution volume of 20 µL, whereas the most efficient amplification kits are THERMO, followed by ELI. The best extraction–amplification procedure appears to be the THERMO (20 µL)–THERMO combination, with the amplification carried on any of the two RT-qPCR devices. Interestingly, this manual extraction–amplification procedure [THERMO (20 µL)–THERMO] is significantly more efficient than the automated NeuMoDx method as well (Figure 2H). The average difference in the Cq values between this manual procedure and the automated equipment for 10 SARS-CoV-2-positive patients was −1.5 (*p* *** = 0.0001).

To validate this conclusion, the SARS-CoV-2 RNA was extracted from nasopharyngeal swab samples collected from three other SARS-CoV-2-positive patients (PA18-–20) with all four extraction kits by using the best yielding elution volumes and further amplified them with the most efficient amplification kits. Once again, THERMO extraction proved to have the highest relative amplification efficiency, followed by ZYMO (Figure 3B). To confirm our results about choosing the best option of extraction–amplification kits, we further used the same experimental approach to process SARS-CoV-2 Reference Material (AccuPlex SARS-CoV-2 Reference Material Kit, #0505-0126, Sera-Care), with known initial RNA concentration (5000 copies/mL). The THERMO (20 µL)-THERMO combination kits represented the best available option (Figure 3C). The last validation step consisted in comparing the efficiency of the best yielding extraction kits, THERMO (in both 20 µL and 50 µL) and ZYMO in 20 µL, on 12 successive dilutions of SARS-CoV-2 Reference Material, amplified with THERMO kit (Figure 3D). Extracting SARS-CoV-2 RNA with THERMO in a final elution volume of 20 µL decreased the Cq values with 1.74 cycles on average (*p* *** < 0.001) compared to THERMO extraction in a volume of 50 µL, and with 2.50 cycles on average (*p* *** < 0.001) in comparison to ZYMO extraction in 20 µL.

### 3.2. The RT-qPCR Limit of Detection of SARS-CoV-2

To determine the RT-qPCR limit of detection of SARS-CoV-2, a serial dilution assay of the SARS-CoV-2 Reference Material (serial dilution from 1:1 to 1:50) was implemented. First, the reliability of the best yielding extraction and amplification kits was tested by evaluating the goodness of fit (*R*^2^) of the linear regression obtained with each extraction–amplification combination for the serial dilutions of the SARS-CoV-2 Reference Material (Figure 4A–D). Amplification with THERMO was found more accurate than that with the ELI kit (Figure 4A vs. Figure 4B). Likewise, the THERMO extraction was superior in accuracy compared to the extraction with ZYMO (Figure 4B vs. Figure 4D). Lastly, the SARS-CoV-2 RNA extraction with the THERMO kit in a final volume of 20 µL had a stronger goodness of fit coefficient than in an elution volume of 50 µL (Figure 4C vs. Figure 4D). Therefore, the THERMO (20 µL)–THERMO combination is the best yielding procedure, as shown earlier, and the most reliable option when analyzed by the expected Cq values according to the initial number of SARS-CoV-2 RNA copies.

In the serial dilution assay, in which the initial number of RNA copies varied between 1000 and 20 copies/extraction, all dilutions were detected by the amplification with THERMO, independent of the extraction kit used (Figure 4F). However, the ZYMO-extracted RNA amplified with ELI was not detected at a dilution of 1:50 (20 copies of initial RNA), proving that for the ELI amplification kit the limit of detection is somewhere between 20 and 25 copies of SARS-CoV-2 RNA before extraction or between 5 and 6 copies of RNA/PCR reaction, under our experimental conditions.

Due to the fact that the limit of detection was not reached for the THERMO amplification by this assay, another serial dilution assay was employed, but this time the SARS-CoV-2 Reference Material was diluted after the extraction (extraction kit: THERMO; elution volume: 20 µL). Three additional dilutions were used in this experiment (1:100, 1:200, and 1:500). The 1:100 dilution, corresponding to 2.5 copies/PCR reaction, was the last concentration detected, at a Cq of 39.8; thus, this might be considered the RT-qPCR limit of detection of SARS-CoV-2 RNA for the THERMO (20 µL)–THERMO combination. Further, we checked the reliability of the extraction–amplification method by comparing the Cq values of standard curves generated from positive control (Amp-PC—TaqMan 2019-nCoV Control Kit v2, #CCU001L, Applied BioSystems by Thermo Fisher Scientific, USA) with both serial dilution assays (before and after extraction) and obtained a high overlapping between these three curves (Figure 4E,F).

### 3.3. The Optimal Pool Size Based on SARS-CoV-2 Prevalence

In our center, the prevalence of SARS-CoV-2 virus was 5.71% among the asymptomatic oncological patients and 0.54% among the tested medical staff.

Even though the Dorfman hierarchical testing approach had the lowest reduction in the expected number of tests, it had the greatest overall sensitivity compared with the other two pooling strategies. Therefore, we chose to test Dorfman hierarchical pooling strategy on our approach. For this pooling method, the calculations predicted an optimal pool size of 5 samples for the asymptomatic oncological patients and 15 samples for the cohort consisting of the medical staff tested in the screening (Table 1).

### 3.4. The Accuracy of the Pooling Strategy of Nasopharyngeal Swab Specimens for SARS-CoV-2 Detection by RT-qPCR

Our previous results demonstrated that the THERMO (20 µL)–THERMO extraction–amplification combination is both the best yielding procedure, with a limit of detection of 2.5 copies/PCR reaction, and the most reliable option among all the combinations tested. Simultaneously, the optimal pool size was found to be between 5 and 15 samples, based on the COVID-19 prevalence registered in our diagnostic center. Therefore, to assess the accuracy of such a pooling strategy, we mixed nasopharyngeal swab samples collected from 15 COVID-19-positive patients with COVID-19-negative samples in ratios of 1:5, 1:10, and 1:15 and further processed them with the THERMO (20 µL)–THERMO extraction–amplification combination. The Cq values of the 15 individual positive samples varied between 24.6 (PA21) and 36.7 (PA35); thus, patients with both high and very low initial viral load were included in this study.

For 12 out of the 15 positive samples included in this study, SARS-CoV-2 was detected by RT-qPCR in all pools in which they were included (1:5, 1:10 and 1:15), as shown in Figure 5A. The Cq values for these 12 samples, when tested individually, were as high as 34.4. To test for the accuracy of the pooling approach, we calculated the average differences observed between the Cq values obtained for individual samples and those registered for pooled samples, and compared them with the expected differences that theoretically should be recorded. Our experimental data strongly overlap with that of the theoretical model, proving the reliability of the pooling strategy under our laboratory conditions (Figure 5B).

The remaining three COVID-19-positive samples, which were characterized by very low viral load (Cq _PA33_ = 35.1; Cq _PA34_ = 35,7; Cq _PA35_ = 36.7), were not detected as positives when pooled with COVID-19-negative samples. Even though some of the SARS-CoV-2 genes targeted by the THERMO kit were amplified in the pooled samples containing these positive specimens (e.g., 1: the N protein in the 1:5 pool containing PA34 sample, Cq = 35.5; e.g., 2: the S protein in the 1:5 pool containing PA35 sample, Cq = 38.2), the pools did not meet the kit’s criteria to be declared positive (at least two out of the three genes amplified), because the other two genes targeted by the amplification kit were not detected at all. Therefore, for these samples, the results are false-negative. However, one might lower the positivity threshold to at least one of three genes amplified instead of two when analyzing pooled samples, while following the kit’s criteria for positivity only afterwards, when testing the individual samples that made up the pool. Several other commercially available, EUA-approved SARS-CoV-2 amplification kits that target multiple sequences consider that the amplification of only one out of the two or three genes tested is enough to meet the positivity criteria [16].

Corroborating all these results, this pooling strategy proves to be efficient and reliable in detecting SARS-CoV-2, but only for samples with Cq values lower than 35 (when tested individually). However, 35 represents a high Cq value, meaning a very low viral load. According to our serial dilution assay for this extraction–amplification procedure, a Cq value greater than 35 is obtained for samples with less than 20 SARS-CoV-2 RNA copies/extraction and less than five copies/PCR reaction (Figure 4F). To assess the frequency by which such false-negative results would appear in our laboratory if this pooling approach would be implemented, we determined the percentage of samples with Cq values higher than 35 out of all the positive samples diagnosed in our diagnostic center during April and May 2020. Out of 438 positive samples, only 26 had Cq values higher than 35; thus the frequency of putative false-negative results under this pooling scheme would be less than 6%. Furthermore, false-negative results would be even less frequent if the positivity threshold for pooled samples would be lowered to at least one of three genes amplified instead of two, as previously described. In this case, samples with a Cq value lower than 35.5 (when tested individually) are still detected in pools consisting of five samples (1:4 ratio). Under these laboratory conditions, only 9 out of the 438 positive samples diagnosed in our diagnostic center would remain undetected, the frequency of false-negative results being 2.05%.

## 4. Discussions

The SARS-CoV-2 pandemic represents the biggest economic, social, and health challenge in the last hundred years. Although several vaccines have been developed quite rapidly, their production and administration to a large population will be possible in years to come. Moreover, due to its RNA instability, the SARS-CoV-2 virus can suffer mutations with high adaptive values, which might promote viral resistance to medication or to the available COVID-19 vaccines [17,18]. Thereby, applying screening in communities with early detection of positive individuals is still one of the most useful approaches in controlling this epidemic. In this way, sample pooling could represent a complementary RT-qPCR option to individual PCR testing for SARS-CoV-2, to increase the testing efficiency [7]. Therefore, this paper presents useful data on developing reliable pooling approaches as a proof-of-concept, demonstrating the reliability of nasopharyngeal swab specimen pooling in detecting SARS-CoV-2. We are confident that developing and implementing such a pooling strategy would strongly enhance the testing efficiency in terms of resources and time, thus greatly increasing the total number of investigated individuals. As we presented above, based on the SARS-CoV-2 prevalence data specific to our diagnostic center, a two-stage hierarchical testing scheme would reduce the expected number of tests by 56–86% (Table 1). Moreover, if we consider that the detection limit of pooling analysis for SARS-Cov-2 analysis was about 2.5 copies per PCR reaction, meaning very sensitive, we can suppose that this method is reliable and very useful in controlling the Covid-19 pandemic. Our results are consistent with previous reports that propose similar sample pooling approaches. For example, Petrovan et al., (2020) proved that pooling of up to 80 samples did not affect the efficacy of the diagnostic assays, if the initial positive sample has a very high (Cq = 16–17) or high (Cq = 25–26) viral load [19]. Nevertheless, in our study we used several samples with much lower initial viral loads (Cq = 30–36) in order to better describe the limits of such a pooling approach.

However, such a sample pooling approach is highly dependent on the infrastructure, reagents, and procedures used in each laboratory. Therefore, this paper is also intended to be a practical guide for SARS-CoV-2 diagnostic centers regarding the optimization steps necessary to be implemented. We suggest that the extraction kits, the final elution volume in the extraction step, the amplification kits, the RT-qPCR instruments, and the overall procedure should be taken into account, as all these parameters influence the results obtained with each diagnostic assay, as shown in this paper. Both the relative efficiency and the accuracy of the assays should be evaluated in order to identify the best diagnostic procedure. Lastly, the optimal pool size should be assessed based on the SARS-CoV-2 prevalence data specific to each region/diagnostic center to maximize the assays’ overall efficiency.

The relatively rapid discovery of the complete genome structure of SARS-CoV-2 facilitated the development of specific kits and laboratory protocols adapted for COVID-19 [20]. Generally, SARS-CoV-2 detection PCR kits aim to analyze several specific regions from genomic RNA, including RNA-dependent RNA polymerase (RdRp) or genes involved in the synthesis of the main structural proteins: spike (S), envelope (E), membrane (M), and nucleocapsid (N). The PCR analysis kits used in our study targeted different viral areas of SARS-CoV-2, as follows: the PD and TIB target the RdRp gene, ELI targets three regions from the N gene, and THERMO targets the S, N, and ORF 1a genes. As we pointed out in our data, both kits, PD and MN, that target the RdRp gene have the lowest detection sensitivity, while the ELI and THERMO kits that target the N gene are more sensitive. It is known that the N protein is abundantly expressed during SARS-CoV-2 infection, being responsible for transcription and replication of viral RNA. Our results indicate that the THERMO kit that targets three genes, including S, N, and ORF 1a, is the most competent for developing a large-scale pooling method. Moreover, the detection of SARS-CoV-2 using S/N proteins is considered more accurate for serodiagnosis methods [21]. Considering these, choosing a PCR kit to analyze at least the N protein and, if possible, the S protein is the best option for developing a large-scale pooling method.

Our data revealed that pooling of nasopharyngeal swab samples proves to be efficient and reliable in detecting SARS-CoV-2, but only for samples with individual Cq values lower than 35, implying the possibility for the occurrence of false-negative results. Nevertheless, the frequency of such samples in our data was lower than 6%, a percentage that might be considered tolerable given the potentially much higher rates of false-negative results due to other errors upstream, such as swab sampling [22,23].

Moreover, if the scale pooling method is applied as a repetitive screening strategy in conjunction with an epidemiological investigation, its sensitivity could become very high, even compared to PCR individual testing. In this regard, a relevant example might be the implementation of this pooling strategy in hospitals as a screening program for both medical staff and inpatients, with a periodic, repetitive investigation, taking into consideration that both healthcare workers and patients have a significantly increased risk of COVID-19 infection [24] and vaccination is not 100% efficient.

In consideration of all this, sample pooling of nasopharyngeal swab specimens is efficient and reliable in detecting SARS-COV-2, saving time and money and increasing the number of investigated people. However, a sample pooling approach is highly dependent on the infrastructure, reagents, and procedures used in each laboratory. Therefore, this paper should serve as a practical guide for SARS-CoV-2 diagnostic centers to determine the most efficient and reliable pooled sample diagnostic assay.

## 5. Conclusions

Our study demonstrated that pooling sample analysis has a very good sensitivity for detecting SARS-CoV-2 if the extraction and amplification steps are appropriately implemented. The success of this method is also dependent on the PCR amplification kit, the high sensibility being related to the investigation of the N gene.

## Figures and Tables

**Figure 1 viruses-13-00902-f001:**
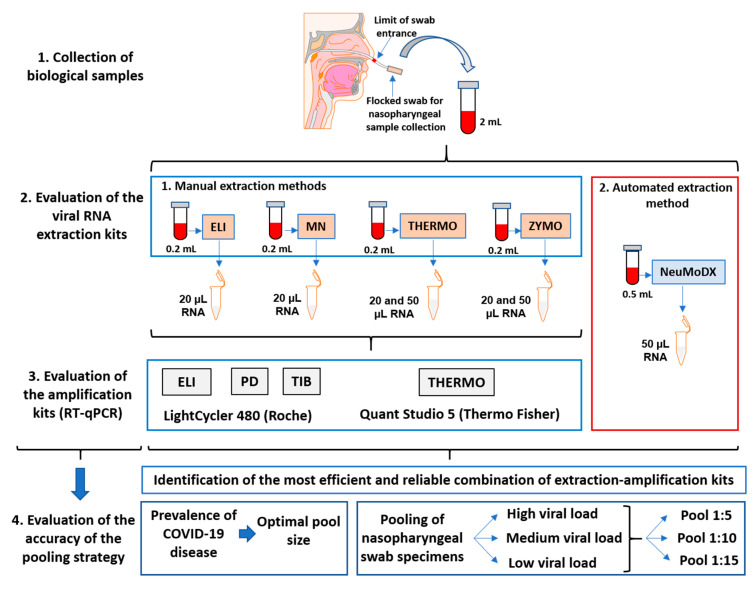
The workflow for large-scale testing of SARS-CoV-2 coronavirus by RT-qPCR using four manual methods for RNA extraction and PCR assessment, two PCR devices, as well as an automated flux for SARS-CoV-2 detection. The symbols for extraction and PCR amplification kits are described in the Materials and Methods section.

**Figure 2 viruses-13-00902-f002:**
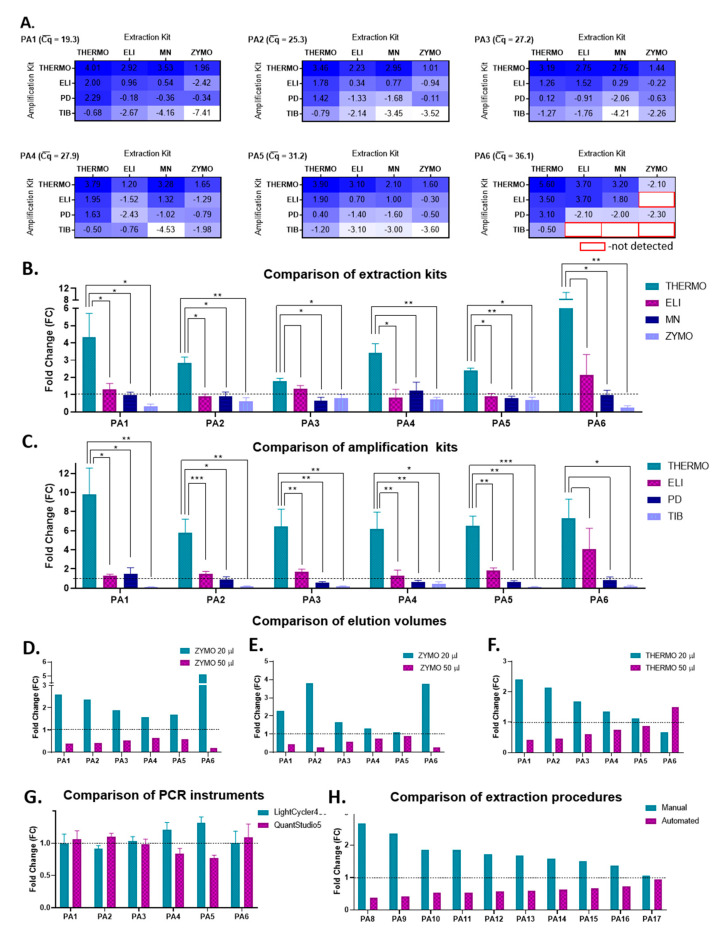
The most efficient combination of extraction and amplification procedures for SARS-CoV-2 detection by RT-qPCR: the relative amplification efficiency for each combination of extraction (THERMO, ELI, MN, and ZYMO) and amplification (THERMO, ELI, PD, and TIB) kits (**A**); the fold change between the extraction kits, without taking into consideration the variances between the amplification kits (**B**); the fold change between the amplification kits, without taking into consideration the variances between the extraction kits (**C**); the fold change between different elution volumes (20 and 50 µL) used in the extraction procedure, for RNA extracted with ZYMO and amplified with ELI (**D**), for RNA extracted with ZYMO and amplified with THERMO (**E**), and for RNA extracted with THERMO and amplified with THERMO (**F**); the fold change between the different RT-qPCR instruments (LightCycler480 and QuantStudio5) for RNA extracted with all four extraction kits (THERMO, ELI, MN, ZYMO) and amplified with ELI, without taking into consideration the variances between the extraction kits (**G**); the fold change between manual (extraction kit: THERMO; elution volume: 20 µL; amplification kit: THERMO) and automated (NeuMoDx) procedures (**H**). The biological samples used for the data presented in (**A**–**G**) were collected from the same six COVID-19-positive patients (PA1–6), whereas samples used for the data presented in (**H**) were collected from 10 other COVID-19-positive patients (PA8-17). Data in (**B**,**C**,**G**) are presented as mean ± SEM, whereas the statistical significance was assessed by a paired *t*-test (*p* * < 0.05, *p* ** < 0.01, *p* *** < 0.001). PA—patient; $\bar{\mathrm{Cq}}$ —average quantification cycle.

**Figure 3 viruses-13-00902-f003:**
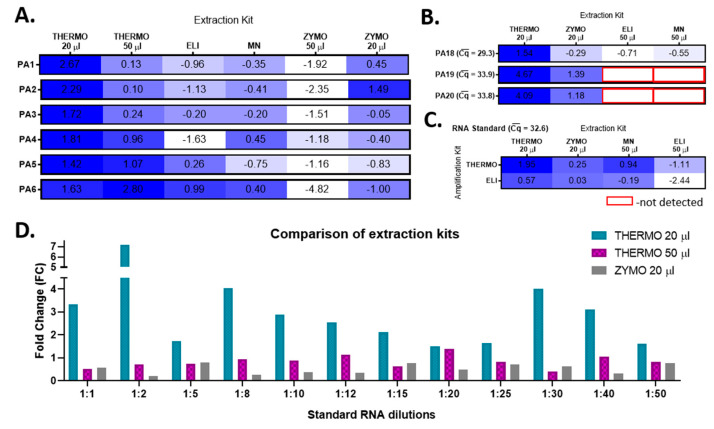
Validation of the most efficient combination of extraction and amplification procedures for SARS-CoV-2 detection by RT-qPCR: the relative amplification efficiency (amplification kit: THERMO) for each extraction kit (THERMO, ELI, MN, and ZYMO), taking into account two different elution volumes for THERMO and ZYMO extraction kits, on specimens from six COVID-19-positive patients (PA1-6) (**A**); the relative amplification efficiency (amplification kit: ELI) for each extraction kit (THERMO, ELI, MN, and ZYMO), using the best yielding elution volumes, on specimens from three other COVID-19-positive patients (PA18-20) (**B**); the relative amplification efficiency (amplification kit: THERMO and ELI) for each extraction kit (THERMO, ELI, MN, and ZYMO), using the best yielding elution volumes, on SARS-CoV-2 Reference Material (**C**); the fold change between the best yielding extraction kits (THERMO and ZYMO) on 12 successive dilutions of SARS-CoV-2 Reference Material (amplification kit: THERMO) (**D**). PA—patient; $\bar{\mathrm{Cq}}$ —average quantification cycle.

**Figure 4 viruses-13-00902-f004:**
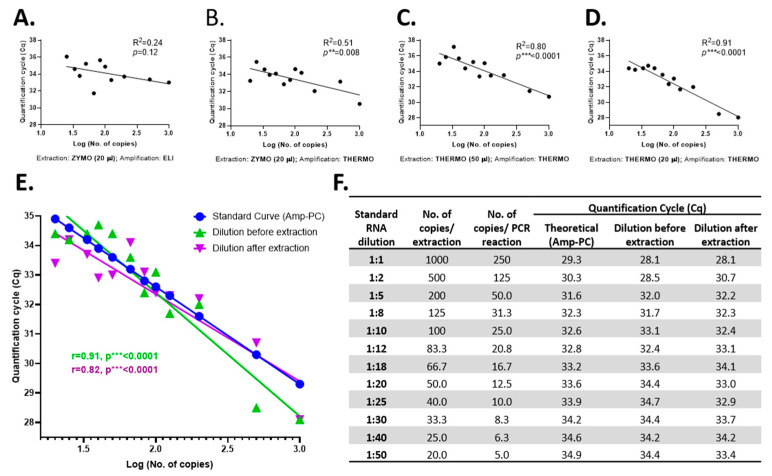
The RT-qPCR limit of detection of SARS-CoV-2 RNA processed with the most efficient combination of extraction and amplification kits: the correlation between the number of RNA copies (serial dilutions) and the quantification cycle (Cq) at which they were detected, for SARS-CoV-2 Reference Material extracted with ZYMO and amplified with ELI (**A**) and THERMO (**B**), extracted with THERMO in a final elution volume of 50 µL (**C**) and 20 µL (**D**) and amplified with THERMO; the overlap between the theoretical standard curve of amplification calculated based on the Cq of the Amp-PC (amplification positive control—TaqMan 2019-nCoV Control Kit v2, #CCU001L, Applied BioSystems) and the regression lines obtained experimentally, based on the amplification of the serial dilutions of SARS-CoV-2 Reference Material, diluted either before or after the extraction step (**E**); the Cq values obtained for each dilution of SARS-CoV-2 Reference Material (before and after extraction) compared with the theoretical Cq calculated based on the amplification of the Amp-PC. For the data presented in (**E**,**F**), the extraction kit used was THERMO (elution volume: 20 µL), and the amplification kit was THERMO. The statistical significance was assessed in terms of the goodness of fit ($R^{2}$) of the linear regression (*p* ** < 0.01, *p* *** < 0.001).

**Figure 5 viruses-13-00902-f005:**
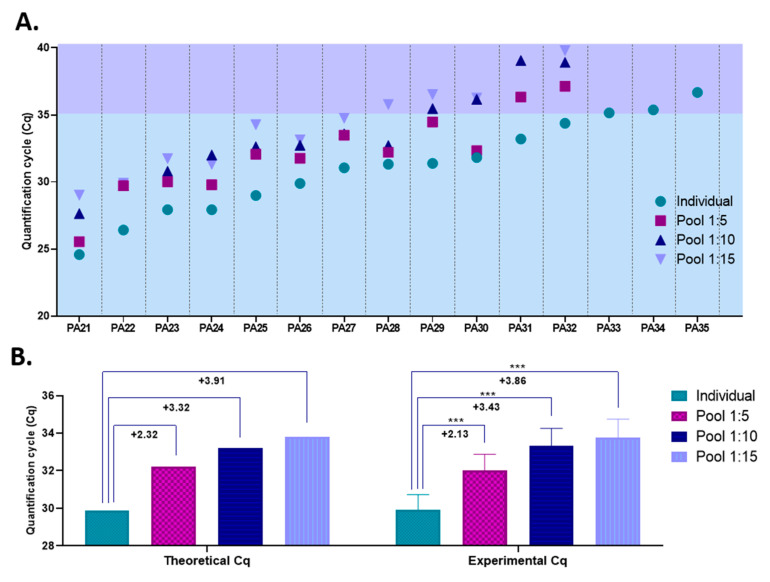
The detection efficiency of SARS-CoV-2 by RT-qPCR in pooled nasopharyngeal swab specimens from patients with COVID-19: the comparative Cq values between individual and pooled nasopharyngeal swab specimens in three different ratios (1:5, 1:10, and 1:15), obtained by mixing one COVID-19-positive sample with 4/9/14 COVID-19-negative samples, for 15 different COVID-19-positive patients (PA21-35) (**A**); comparison between the expected (theoretical) and experimentally obtained differences between the Cq values at which individual and pooled nasopharyngeal swab specimens were detected (**B**). In (**B**), (experimental Cq) data are presented as mean ± SEM of Cq values of all 15 individual or pooled samples, whereas the statistical significance was assessed by a paired *t*-test (*p* *** < 0.001).

**Table 1 viruses-13-00902-t001:** The optimal pool size, the overall sensitivity, and the reduction in the expected number of tests in three different pooling strategies, based on the prevalence of COVID-19 disease observed in a cohort of asymptomatic oncological patients (prevalence: 5.71%; *n* = 471) and among asymptomatic medical staff (prevalence: 0.54%; *n* = 1117).

Pooling Strategy	Prevalence (%)	Optimal Pool Size (Samples) *	Overall Sensitivity *	Reduction in the Expected No. of Tests *
Dorfman hierarchical testing (2 stages)	0.54%	15	0.902	86%
	5.71%	5	0.902	56%
Sterrett hierarchical testing (3 stages)	0.54%	36-6-1	0.857	92%
	5.71%	9-3-1	0.857	61%
Array testing	0.54%	<20 × 20	0.870	89%
	5.71%	10 × 10	0.858	60%

* All calculations were performed in “The Shiny” application for pooled testing, available at https://www.chrisbilder.com/shiny.

## Data Availability

Data sharing not applicable.

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
