# Peer review of "Critical Aspects Concerning the Development of a Pooling Approach for SARS-CoV-2 Diagnosis Using Large-Scale PCR Testing"

_viruses, 2021, doi:10.3390/v13050902_

Round 1

Reviewer 1 Report

The manuscript “Critical aspects concerning the development of a pooling approach for SARS-CoV-2 diagnosis using large-scale PCR testing” by Cruceriu and colleagues is a well-designed, thoughtfully executed and successfully written piece of work that carefully documents and shares the experience of the group behind the Romanian national screening program in assessing kits, platforms, strategies and approaches to SARS-CoV-2 test pooling. Overall, this work offers great value to the readership and could inform the development of pooling approaches elsewhere, which remains an important public health need, especially, as the authors indicate, in places such as schools or hospitals. The level of detail for materials, methods, experimental set up, introduction and discussion, is very good. The generally transparent language, educational approach, and lack of overinterpretation or overgeneralization is refreshing. 

Perhaps the largest drawback of the study is the relatively small sample size. However, given the focus on evaluating multiple extraction, amplification and testing platforms, which adds great complexity to the experimental set up, and given that existing samples do cover a wide range of expected viral loads and include reference materialsthis small number is sufficient to deliver the message presented by the authors. Extension of the conclusions reached here to a larger, independent validation cohort would be of great interest but is admittedly outside the scope of the present work, which is valuable in its current form. 

The main piece of feedback I would offer to potentially improve this study concerns the criteria for positive interpretation of pooled SARS-CoV-2 tests as stated in the paragraph spanning lines 363-370. While it is important to follow the kit’s criteria for positivity when testing individual samples (at least 2 of out the 3 genes amplified, as stated), it would be helpful to consider and discuss lowering the pooled positivity threshold to at least 1 of 3 genes amplified instead of 2. Then, once the pool is subsequently assessed as individual samples, the standard 2 out of 3 targets can be used for declaring positivity. Based on the authors’ results and discussion, this strategy would lower the false negative rate without significantly compromising on potential false positives. Given that the societal cost of false negatives is more severe than that of false positives, having a more lenient calling strategy for pooled samples, especially while maintaining the standard calling strategy for the individual sample follow-up, would amplify the value of pooling strategies. 

Therefore, it would be helpful if authors evaluated their false positive and negative rates considering this alternative strategy for calling positivity within pools and addressing their experience with this alternative strategy in the discussion. 

A helpful reference in this context is PMID 33401392 “Nucleic Acid-Based Diagnostic Tests for the Detection SARS-CoV-2: An Update”. Within their supplementary table, the authors list the positivity criteria for many existing, EUA-approved SARS-CoV-2 tests, which can serve to justify the validity of, and precedent for, different criteria, even if not explicitly in the context of a pooling approach. 

Author Response

Dear Reviewer,

Please find uploaded our response to your requests!

Best regards!

Ovidiu Balacescu

Reviewer 2 Report

The authors are thoroughly described the experimental design used in their lab using the pooling approach to improve the capacity for SARS-CoV-2 diagnostics. Although this is not a novel idea, moreover some references were omitted by the authors regarding pooling design validation. However, there is still need for proper scientific information that local healthcare authorities can use, to increase the diagnostic capacity. Since the authors are presenting a highly detailed image of the workflow, I will encourage to make some necessary modifications:

Please rewrite parts of the introduction in order to include the Romanian response to the pandemic in terms of diagnostic capacity, different approaches, number of tests etc. Also please avoid repetitions in part of the introduction and remove parts that are not directly involved to the current research. I would strongly encourage the authors to have a native speaker proofread the manuscript. 

Please mention the work described from other researchers from Romania, comparing your results and draw conclusions (https://doi.org/10.3390/diagnostics10070472 )

line 62: please add real time qPCR

line 65: change "harvesting" with "collecting"

lines 69-76: this paragraph needs to be removed since it's not relevant to the current research. 

lines 77-80: please avoid repetitions

lines 81-98: this part is confusing. It is already well documented the fact that the sample pooling approach can be used in order to increase the number of tests done for COVID and other diseases. Also it is well known how the pooling strategy works. 

MM: Did you perform the sample analysis (extraction and pooling) in duplicate, or triplicate? 

lines 103-104: It is not clear for the sentence what are the authors trying to present: overview on the number of tests? All the tests were pooled (before or after extraction). Please re-write the sentence

lines 127-142: can you describe briefly how the extraction procedure was performed for all the kits used?

Figure 1: it is not clear where the samples were pooled.

line 143: what are the PCR parameters? Were optimized based on the manufacturers instructions? What is the Ct threshold for every kit?

Figure 2: This experiment should have been done with an internal control, then as comparison a known positive sample should have been used. Were all the samples extracted and analyzed at the same time? Please add that to the text

Figure 5A: for which gene is this data set presented for?

lines 362-370: How about the other genes that the kits are amplifying? 

line 397: mutations, instead of mutation. Please rephrase "adapts it to new survival conditions"

line 453-455: repetition. please delete this sentence

Author Response

(The authors gave the same response as above.)

Round 2

Reviewer 2 Report

The authors made all the necessary modifications, therefore only minor spelling and modifications should be done.